# Support or Risk? Software Project Risk Assessment Model Based on Rough Set Theory and Backpropagation Neural Network

**Xiaoqing Li [1], Qingquan Jiang [1,\*] , Maxwell K. Hsu [2] and Qinglan Chen [1,\*]**

1    School of Economics & Management, Xiamen University of Technology, Xiamen 361024, China
2    Marketing, University of Wisconsin-Whitewater, Whitewater, WI 53190, USA
\*    Correspondence: jiangqingquan@xmut.edu.cn (Q.J.); chenqinglan@xmut.edu.cn (Q.C.)

**Abstract:** Software supports continuous economic growth but has risks of uncertainty. In order to improve the risk-assessing accuracy of software project development, this paper proposes an assessment model based on the combination of backpropagation neural network (BPNN) and rough set theory (RST). First, a risk list with 35 risk factors were grouped into six risk categories via the brainstorming method and the original sample data set was constructed according to the initial risk list. Subsequently, an attribute reduction algorithm of the rough set was used to eliminate the redundancy attributes from the original sample dataset. The input factors of the software project risk assessment model could be reduced from thirty-five to twelve by the attribute reduction. Finally, the refined sample data subset was used to train the BPNN and the test sample data subset was used to verify the trained BPNN. The test results showed that the proposed joint model could achieve a better assessment than the model based only on the BPNN.

**Keywords:** Backpropagation neural network; risk assessment; rough set theory; software projects risk

## 1. Introduction

Although the software industry has contributed significantly to the economic growth in many countries, unfortunately, the development of many software projects could not be considered successful [1,2]. The CHAOS report conducted by the Sandish Group, which included more than 50,000 software projects around the world, revealed that the mean success rate of software development projects during the 2011–2015 period was 30% at most (Table 1). In other words, more than 70% of software projects would be categorized as challenging or failed [3].

**Table 1.** Standish Group survey on global software projects in the period 2011–2015.

|                     | 2011 | 2012 | 2013 | 2014 | 2015 |
| ------------------- | ---- | ---- | ---- | ---- | ---- |
| Successful projects | 29%  | 27%  | 31%  | 28%  | 29%  |
| Challenged projects | 49%  | 56%  | 50%  | 55%  | 52%  |
| Failed projects     | 22%  | 17%  | 19%  | 17%  | 19%  |

Serious problems exist in assessing future risks across a broad cross-section of industries [2]. In the late 1980s, software risk management was introduced into the area of software project management for the first time by Barry Boehm, who is considered a notable pioneer in this research field. Boehm [4] believed that identifying and dealing with the risks in the early development stage could lessen long-term costs and help prevent software failures. According to IEEE research in the late 1990s, 50–70% of the risks could have been found through the project analysis, while 90% of them could

have been avoided [5]. Therefore, assessing and predicting the risks in the early stage of software project development are essential to manage the risks and improve the success rate of software development projects.

Nowadays, enterprises have an increasing dependence on information technology and many enterprises develop or customize their application software. The downside is that an unsuccessful software development project is likely to lead to a big loss for enterprises. Thus, enterprises should evaluate and analyze the risks of software projects to be planned by identifying the potential risks more accurately and adopting scientific methods for risk mitigation. Software risk factors are various and complicated, and historical data are characterized by uncertainty and are unstructured; therefore, the models or algorithms that require prior knowledge are not applicable. In recent years, the risk assessment methods that explore accurate value and information from a large amount of incomplete, inaccurate, and fuzzy data have been considered. This paper proposes a risk assessment model that integrates a backpropagation (BP) neural network (BPNN) with the rough set. It takes advantages of both the BP neural network and the rough set to improve the accuracy of risk assessment.

This paper is organized as follows. Section 2 describes the related work. Section 3 provides a brief overview of the related methods and approaches. Section 4 identifies the software projects risk factors and constructs the software project risk assessment model combining the rough sets and BP neural network. Section 5 describes the implementation details, data collection process, the experiments and results. Lastly, Section 6 provides conclusions and guidelines for our future work.

## 2. Related Work

The Webster dictionary defines "risk" as "the possibility of loss or injury." Software project risk has been defined as a product of uncertainty associated with the project risk factors and the magnitude of the potential loss due to project failure [6]. Boehm classified the software risk management into two parts, risk assessment and risk control. As a primary step of risk management, risk assessment involves risk identification, risk analysis, and risk prioritization [4]. The critical project examination by using different risk assessment methods helps researchers and practitioners to evaluate the impact of various project-related risks on project success. Numerous machine learning and data mining algorithms have been used in risk analysis, including artificial neural networks [7], Bayesian belief network [8], and discriminant analysis [9], etc. The artificial neural networks (ANNs) are commonly applied to achieve better learning and analytic abilities in solving sophisticated software project risk assessment problems than the traditional methods [10]. The ANNs do not require the establishment of relations and a conditional probability table. Thus, when it is difficult to build the relationship between software project risks and outcomes, neural networks denote an effective solution. Therefore, neural networks represent a good candidate for establishing a project risk assessment model. However, ANNs have the disadvantages that an inappropriate ANN structure, which is often determined subjectively, may result in poor training efficiency and network performance. In recent research, ANNs have been combined with many other approaches to improve the software risk assessment. Neumann [11] aimed to treat the software risk analysis from the objective standpoint; namely, he put forward a technique, which combines principal component analysis and ANNs to analyze the software risk. Hu et al. [12] employed ANNs and a support vector machine to establish a model for software project risk assessment. Hu et al. [13] also proposed a model using the Bayesian networks with causality constraints for risk analysis of software project development. Goyal et al. [2] integrated an ANN with fuzzy logic to form a neuro-fuzzy technique for software project risk assessment.

As a promising method to deal with inaccurate, uncertain, and incomplete data, the rough set theory is introduced in this paper. In computer science, the rough set theory represents a new achievement in data mining technology, which was first introduced by the Polish scientist Z. Pawlak [14]. Unlike the fuzzy set or Bayesian theory that requires membership functions and prior knowledge, the rough set theory could disclose hidden knowledge, reveal potential rules, and reduce attributes without any additional information or statistical assumption but their datasets [15]. The rough set theory can

also effectively avoid personal subjective influence in the process of information mining. Furthermore, it can process imprecise, inconsistent, incomplete information, and eliminate redundant data and noise data by attribute reduction and value reduction, reducing the data dimension. However, in practice, the rough set theory exhibits poor generalization ability and sensitivity to noise. On the other hand, BP neural networks, as nonlinear self-learning algorithms, show good anti-noise ability by modifying the network weights and thresholds to adjust the network accuracy [16]. However, BP neural networks cannot determine redundant information and reduce the space dimension. Namely, massive data input will cause a complex network structure and increase time consumption [17]. The two mentioned algorithms can be combined to gain complementary advantages to improve the information input, reduce noise interference, and improve network training efficiency.

In the complicated and changeable internal and external environment, the project risk is characterized by diversity and high uncertainty. The limitation of a single project risk identification and assessment method is obvious. Therefore, this study combines the rough set and BP neural network to establish a software project risk assessment model to deal with uncertain, incomplete, and imprecise risk information. First, as a front-end processor of a BP neural network, the rough set reduces redundant data and attributes of a software project risk dataset and refines the key risk factors to simplify the input data of BP network. Then, the network models are constructed to be trained and tested with the training samples and testing samples, respectively, which have been refined and simplified by the rough sets in order to get a reliable assessment model for evaluating and predicting software project risks, which can facilitate the management's decision-making task.

## 3. Methodology

### 3.1. Rough Set Theory (RST)

As a new data mining technology, the rough set theory represents a data analysis tool, which is able to analyze and deal with inconsistent, inaccurate, and incomplete information effectively. It can reduce redundant information and discover hidden knowledge and potential rules while retaining key information. The basic concepts of the rough set are as follows:

#### 3.1.1. Knowledge System (Information System)

A knowledge system can be expressed as $S = \langle U, A, V, f \rangle$, where $U$ denotes a finite set of objects, called the universe, $A = C \cup D$ denotes an attribute set, which consists of the condition attribute subset $C$ and decision attribute $D$, and $C \cap D = \varnothing$. $V = \cup p \in AV_p$, $V_p$ denotes the set of values assumed by the attribute $p$ (called also the domain of attribute $p$), and $f : U \times A \rightarrow V$ is an information function, where $f(p, q) \in V$ for every $p \in U, q \in A$. Therefore, a form datasheet of an information system $S$ is sometimes referred to as a decision table.

#### 3.1.2. Indiscernibility Relation.

For every attribute set $P \subseteq A$ and two objects $X, Y \subset U$, $X$ and $Y$ are indiscernible if and only if $f(X, a) = f(Y, a)(a \in P)$. An indiscernible relationship is defined as:

$$ind(P) = \{(X, Y) \subset U; \forall a \in P, f(X, a) = f(Y, a)\} \tag{1}$$

#### 3.1.3. Upper Approximation and Lower Approximation

Suppose that $U$ is a given universe and $R$ is a family of equivalence relations of $U$, then $K = (U, R)$ denotes a given knowledge base. For every set $X \subset U$, $X$ is called a definable when it can be represented as a union in some $R$; otherwise, $X$ is called an indefinable. The definable set of $R$ is called the

precise set while the indefinable set of *R* is called the rough set, namely, they are the upper and lower approximation of *X*, respectively, which is expressed by:

$$R^-(X) = \cup\left\{Y \in U \middle| ind(R) : Y \cap X \neq \varnothing\right\} \tag{2}$$

$$R\_(X) = \cup\left\{Y \in U \middle| ind(R) : Y \subseteq X\right\} \tag{3}$$

$$POS_R(X) = R\_(X) \tag{4}$$

where $POS_R(X)$ is called the R-positive region of *X*.

### 3.1.4. Attribute reduction (knowledge reduction)

For a knowledge system $S = \langle U, A, V, f \rangle$, attribute set $Q \subseteq C$ denotes a reduction of the condition attribute set *C*, if $POS_Q(D) = POS_C(D)$ and each attribute from *Q* is indispensable to the decision attribute set *D*.

### 3.2. Backpropagation Neural Network

The backpropagation (BP) neural network represents a multi-layer feedforward neural network whose main features are signal forward transmission and error feedback propagation. When transmitting signals forward, the network processes the input signal from the input layer to the output layer through the hidden layer. Each network layer consists of neurons and affects only the next layer. If the output layer yields an unwanted output, the network performs the feedback propagation of the information about the error to adjust the weights and thresholds of the network according to the prediction error, so that the BP neural network prediction output is closer to the expectation [16]. The structure of a typical three-layer BP neural network is shown in Figure 1.

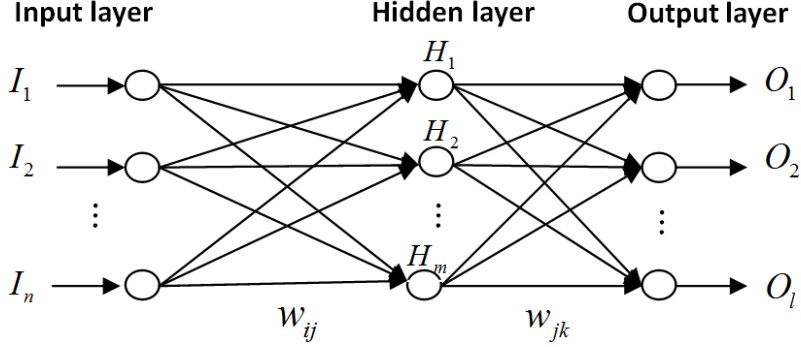

**Figure 1.** The structure of a typical three-layer backpropagation (BP) neural network.

As shown in Figure 1, a three-layer BP neural network is structured as a combination of three successive layers, the processing neurons and interconnection networks. The three layers are the input layer (I)), hidden layer (H)), and output layer (O)). The neurons in each layer have an activation function. The activation function of the neurons within one layer is the same. Each connection between the input layer and the hidden layer is assigned to a weight W and each neuron is associated with a threshold V. The mapping relationships between layers are expressed as:

$$H_j = f_1(\sum_{i=1}^{n} w_{ij}I_i - v_j), j = 1, 2, \cdots, m \tag{5}$$

$$O_k = f_2(\sum_{j=1}^{m} w_{jk}H_j - v_k), k = 1, 2, \cdots, l \tag{6}$$



where $I_i$ denotes the input $i$ in the input layer, $H_j$ denotes the output $j$ in the hidden layer, and $O_k$ denotes the output k in the output layer. $W_{ij}$ represents the neuron connection weight between input $i$ and output $j$, and $Wjk$ represents the neuron connection weight between input $j$ and output $k$. and denote the thresholds of the hidden layer and output layer, respectively, and denote the number of neurons in the hidden layer and output layer, respectively.

In the output layer, actual output $O_k$ is compared with the desired output $E_k$, and the error $C_k$ is then calculated by:

$$C_k = E_k - O_k, k = 1, 2, \cdots, l \tag{7}$$

Therefore, the weight $W$ and threshold $V$ are adjusted when the error is propagated back using a standard learning algorithm. The learning (i.e., the training process) ends when the error and training epochs reach the predefined values.

$$w_{ij} = w_{ij} + \eta H_j\big(1 - H_j\big)I_i \sum_{k=1}^{l} w_{jk}C_k \tag{8}$$

$$w_{jk} = w_{jk} + \eta H_j C_k \tag{9}$$

$$v_j = v_j + \eta H_j\big(1 - H_j\big)I_i \sum_{k=1}^{l} w_{jk}C_k \tag{10}$$

$$v_k = v_k + \eta C_k \tag{11}$$

where $\eta$ is the learning rate and it is between 0 and 1.

## 4. Modeling

### 4.1. Risk Factors Identification

Risk factors identification is the element task of risk assessment. It refers to the process of judging and classifying the present and/or potential risk sources or risk factors, as well as identifying the risk property. Brainstorming is a frequently-used method for risk identification [18]. In this study, brainstorming is mainly used for the identification of software project risk factors. First, based on the literature review, we use the methods of project research and an expert interview to identify various risk factors and risk sources. Then, the identified risk events are summarized and finally, the initial list of software projects risk factors is established. On the one hand, this work refers to the Boehm model [4], the software projects risk classification method proposed by the SEI (Software Engineering Institution) and research achievements of the authoritative software projects risk identification [19–21]. On the other hand, 35 software project experts were invited for interviews and questionnaires. With the support and help of the Xiamen Economic & Information Bureau and Xiamen Software Park Management Committee, 35 experts were selected by recommendation and the snowball sampling method. These experts included software project managers, software engineers, and professors of software engineering, all of whom have the experience of software project development. In the interviews, the brainstorming method was adopted. By putting heads together, the risk factors and risk sources were identified, and the identified risk factors and events were summarized and classified using the affinity diagram method to form the initial risk list, as shown in Table 2. This risk list included two aspects, project risks identification and project results assessment. The project risks included 35 items of risk factor description ($c1 \sim c35$), which were grouped into six categories as the project requirements, project technologies, project management, project team, customers, and environmental complexity.

**Table 2.** Final survey of software project risks.

| Risk Category | Risk Factor and Corresponding Abbreviation |
|---|---|
| Project requirement | Continual system requirements changing (c1); inadequately identified system requirements (c2); unclear system requirements (c3); incorrect system requirements (c4). |
| Project technology | Project involves the usage of new technology (c5); high-level of technology complexity (c6); immature technology (c7); poor scalability of old system (c8); inadequate estimation of required resources (c9); inappropriate software technical design (c10); lack of mechanism for product validation and verification (c11). |
| Project management | Lack of top management commitment (c12); lack of an effective project management methodology (c13); improper change managing (c14); poor project planning (c15); poor project control (c16); inexperienced project manager (c17); ineffective communication (c18); project milestones unclearly defined (c19); corporate politics with negative effect on project (c20); |
| Project team | Inexperienced team members (c21); team members lack specialized skills required by the project (c22); insufficient/inappropriate staff (c23); staff volatility (c24); insufficient support form manager (c25); inadequate training of project team (c26). |
| Project user | Failure to gain user commitment (c27); lack of adequate user involvement (c28); conflict of interest in key sectors of the customer department (c29); low level of customer IT infrastructure (c30); user resistance to change (c31) |
| Environment complexity | Stability of customer environment (c32); introduction of new technology (c33); change in project scope or resource (c34); complexity and chaos in the operation flow (c35) |

*4.2. Software Project Risk Assessment Model*

In the present study, the rough set and BP neural network are combined to construct a software project risk assessment model to monitor the software projects risks. The core idea of this joint model is as follows. First, the rough set is used to perform attribute reduction of the sample data of software project risk factors; then, the reduction set is fed to the input of a BP neural network for training to obtain the mature classification model; finally, the model output is used to evaluate the software project risk level and help decision-makers achieve better decision-making outcomes. The specific process is presented in Figure 2, and the specific steps are as follows:

(1) A project manager evaluates the developed projects according to the initial list of risks, collects the historical data, and obtains the sample set of project risks. The sample set is divided into two sample sets, one intended for learning and the other intended for testing.

(2) Once the condition attributes and risk attributes are defined, sample sets are input to ROSETTA, which is a rough set software, where the discretization and normalization pretreatments are performed. Using the rough set algorithm, the attribute reduction and value reduction are performed, resulting in a simplified sample set.

(3) After the initialization of the structure and parameters of a BP neural network model, the simplified learning sample set of risks is input to the MATLAB neural network toolkit. Through the learning (training) process based on the signal forward-propagation and error back-propagation, the optimal network parameters are obtained. Afterward, the test sample set is adopted to verify the assessment accuracy of the developed neural network model.

(4) The developed model is applied to the risk assessment of practical software projects to obtain the risk assessment report, which provides a reference frame for the subsequent risk control.

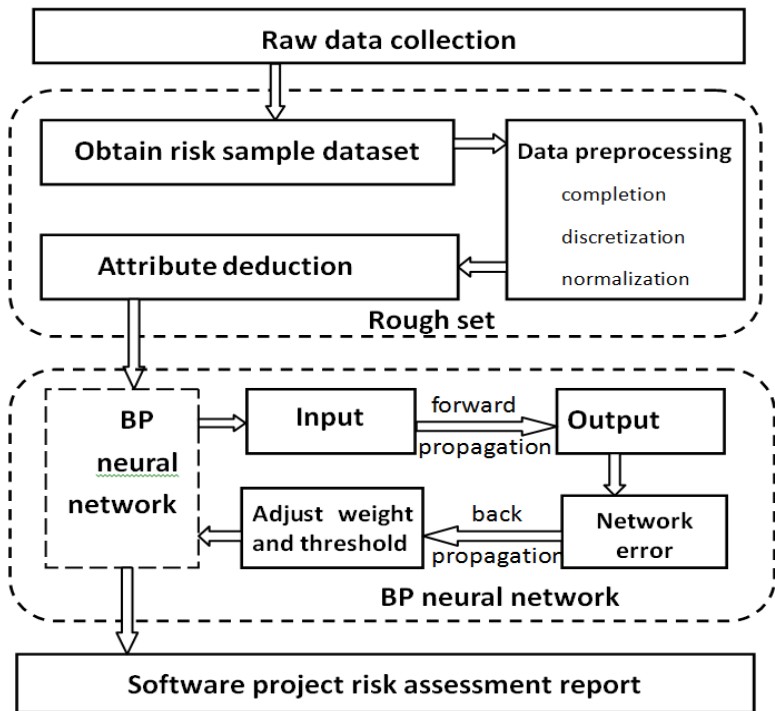

**Figure 2.** The model for software project risk assessment based on the RST and BPNN.

## 5. Experiments and Results

### 5.1. Data Collection

In this study, a questionnaire with a focus on the risk factor items was designed for a single project (see Table 2). The questionnaire was distributed to 35 experts on software projects. With regard to one or two delivered software projects in which these 35 experts were familiar with. The experts were required to evaluate the involved risk factors according to the development process and historical documents; 52 questionnaires were reclaimed. On the other hand, the online questionnaires were distributed to software project managers or software development engineers, and 39 questionnaires were reclaimed. Ninety-one questionnaires were obtained based on the real software projects from 2014 to 2017, including the smartphone application (APP) software development projects in Xiamen.

In view of the evaluation of project risk factors, the respondents had to evaluate both the possibility of risk occurrence and the severity of consequence. The researchers calculated the scores of risk factors evaluation after reclaiming the questionnaires. The risk factor evaluation was performed based on the risk effect (RE). The RE was calculated as (Boehm,1991): RE = Prob(Loss) x Size(Loss), where Prob(Loss) denoted the possibility of risk occurrence, and its value was in the range [0, 1], and Size(Loss) denoted the severity of risk occurrence. This was expressed by a discrete score, which was in the range from 1 to 9, wherein 1 represented the lowest risk and 9 represented the highest risk. Therefore, the value of RE was in the range [0, 9]; when the RE was in the range [0, 3], the risk was low; when it was in the range (3, 6], the risk was medium; and when it was in the range (6, 9], the risk was high. Eventually, the respondents were asked to wholly evaluate the observed projects in order to determine the risk level D of the whole project as low risk (successful project), medium risk (challenging project), or high risk (failed project). The obtained results are given in Table 3.

**Table 3.** Evaluation of software project risk factors.

| Risk Factors | Probability(Loss) | Size (Loss) | Risk Exposure | Risk Level |
|---|---|---|---|---|
| C1 | 20% | 6 | 1.2 | low |
| C2 | 80% | 9 | 7.2 | high |
| C3 | 60% | 6 | 3.6 | medium |
| C4 | 10% | 8 | 0.8 | low |

*5.2. Data Preprocessing and Attribute Deduction*

In this study, the software project risk dataset was obtained through questionnaires. The universe of the domain set $U = \{u_1, u_2, \cdots, u_{91}\}$ and condition attribute $C = \{c_1, c_2, \cdots, c_{41}\}$ were established, and D represented the decision attribute, which was determined by the risk level after the project was accomplished.

As the risk factor data were obtained from different projects by different companies, and part of the data may have been lost to some extent. Therefore, the original data was not complete and had to be patched. Since the rough set theory can analyze only discrete data, the assessment scores were discretized and normalized. Also, the rough set cannot dispose of continuous data, so the equidistant mathematical discrete method was adopted to classify the risks into three levels based on the RE value: When the RE value was in the range [0, 3], the risk was classified as a low risk, when the RE value was in the range (3, 6], the risk was classified as a medium risk, and when the RE value was in the range (6, 9], the risk was classified as a high risk. For the convenience of calculation, the descriptive text was converted into a number: High risk was denoted as 2, medium risk was denoted as 1, and low risk was denoted as 0. Thus, in the decision attribute assignment, numbers 0, 1, 2 represented the low risk (successful project), medium risk (challenged project), and high risk (failed project), respectively.

After obtaining the assessment and decision table of software project risk factors, the rough set theory was adopted for risk attributes reduction, and the virtue of the rough set software ROSETTA, developed collaboratively by the Norwegian University of Science and Technology and Warsaw University was used [22]. The frequently-used reduction methods, the genetic algorithm and Johnson's algorithm, were adopted for learning sample data reduction. According to the two parameters, the support degree and reduction subset length, the optimal reduction set was selected:

$$C^* = \{c_1, c_3, c_4, c_{11}, c_{14}, c_{16}, c_{18}, c_{25}, c_{28}, c_{31}, c_{33}, c_{39}\} \tag{12}$$

*5.3. BP Neural Network Structure Initialization and Training*

In this work, the traditional three-layer BP neural network structure was adopted. According to the number of indicator factors in the above-mentioned optimal reduction set, the number of input-layer neurons of the BP neural network was set to 12. Since the decision attributes of the above-mentioned rough set, which corresponded to the software project risks, were classified into three levels, the number of output-layer neurons was set to 3. The output value was expressed as a binary value. That is, the output $\begin{bmatrix} 1 & 0 & 0 \end{bmatrix}$ represented a high risk, the output $\begin{bmatrix} 0 & 1 & 0 \end{bmatrix}$ represented a medium risk, and the output $\begin{bmatrix} 0 & 0 & 1 \end{bmatrix}$ represented a low risk. The number of hidden neurons was determined according to the following empirical formula: $m = \sqrt{n+l} + a$, where $m$ denoted the number of hidden neurons, $n$ denoted the number of input neurons, $l$ denoted the number of output neurons, and $a$ denoted a constant, and $a \in [1, 10]$; also, $m < n - 1$, thus $m \in [5, 11]$. Namely, the cut-and-try method was adopted to determine the value of $m$. With the aim to determine the optimal number of hidden neurons, different models were established. The number of hidden neurons that corresponded to the least network error was selected as the value of $l$. The relationship between the number of hidden neurons and network error is presented in Figure 3, wherein it can be seen that when $m = 9$, the mean square error (MSE) value was on its minimum.

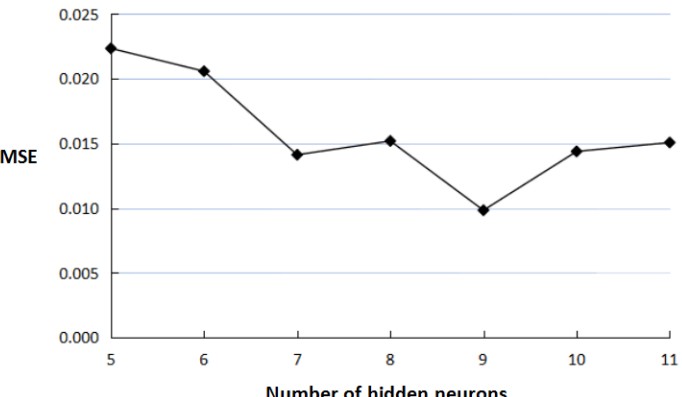

**Figure 3.** The mean square error (MSE) for a different number of hidden neurons.

After the number of layers in the network model and the number of neurons in each layer were determined, the network training parameters were defined. Then, given the network structure, weights and thresholds, we compared the mean square errors of the different combinations of transfer functions, and found the best combination. The function tansig was selected as an activation function of the hidden layer, purelin as an activation function of the input layer; trainlm was selected as a training function. The maximum training epochs were set to 10,000, the value of the MSE to 0.01, and the learning rate to 0.01.

In this work, 81 simplified learning sample subsets were extracted after rough set reduction, and the MATLAB neural network toolkit was used for BP neural network training. When the number of iterations reached the value of 9, the mean square error of the network was 0.0099, which meant the predefined goal was achieved, and at that moment, the network performance was the best (Figure 4).

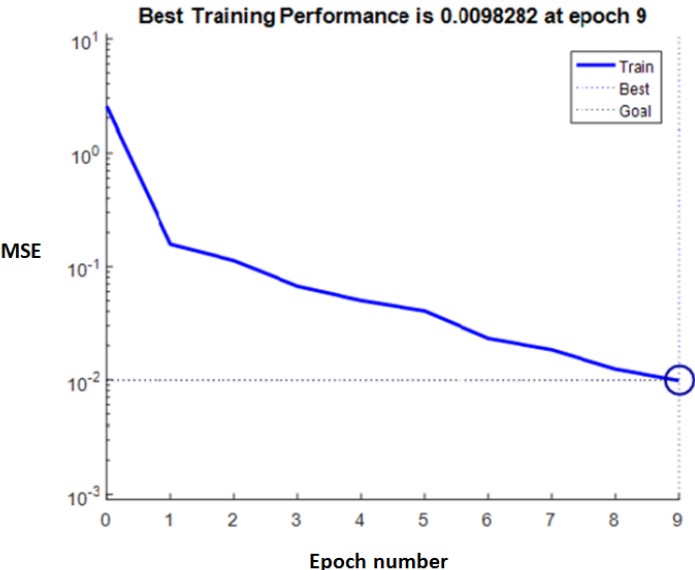

**Figure 4.** RST-BP neural network training performance.

Next, the trained neural network was used for prediction and verification. The test sample set included the risk data of 10 projects. The prediction results obtained by the neural network model were compared with the actual values. The comparison results are shown in Table 4, wherein it can be seen that prediction accuracy was 100%. Therefore, the proposed model showed excellent classification performance.

**Table 4.** Comparison of the predicted and actual output values.

| Project | Network Output | | | Expected Output | | | Output Signal Matching Achieved |
|---|---|---|---|---|---|---|---|
| 1 | 0 | 1 | 0 | 0.1291 | 0.8142 | 0.0561 | yes |
| 2 | 0 | 0 | 1 | 0.0616 | −0.0458 | 0.9847 | yes |
| 3 | 1 | 0 | 0 | 1.0309 | −0.0650 | 0.0346 | yes |
| 4 | 0 | 1 | 0 | 0.0931 | 0.7958 | 0.1113 | yes |
| 5 | 0 | 0 | 1 | −0.0236 | −0.0746 | 1.0981 | yes |
| 6 | 0 | 0 | 1 | −0.0803 | −0.0622 | 1.1425 | yes |
| 7 | 1 | 0 | 0 | 0.9666 | −0.0191 | 0.0522 | yes |
| 8 | 0 | 1 | 0 | −0.0456 | 0.9930 | 0.0523 | yes |
| 9 | 0 | 0 | 1 | 0.0412 | −0.0369 | 0.9966 | yes |
| 10 | 1 | 0 | 0 | 0.8844 | 0.0717 | 0.0439 | yes |

Next, we analyzed the case where only the BP neural network model without attribute reduction by the rough set was used to predict the whole risk of the project, where 35 risk factors in the sample set were all regarded as the network input (Figure 5). Therefore, the number of neurons in the input layer was set to 35, the number of hidden neurons was set to 9, and the number of output neurons was set to 3. The training parameters were kept unchanged. After 48 iterations, the model reached the goal regarding the predefined MSE value 0.01. However, this model had a larger prediction error than the previous model and the accuracy on the test set was only 50%. The training results are shown in Table 5.

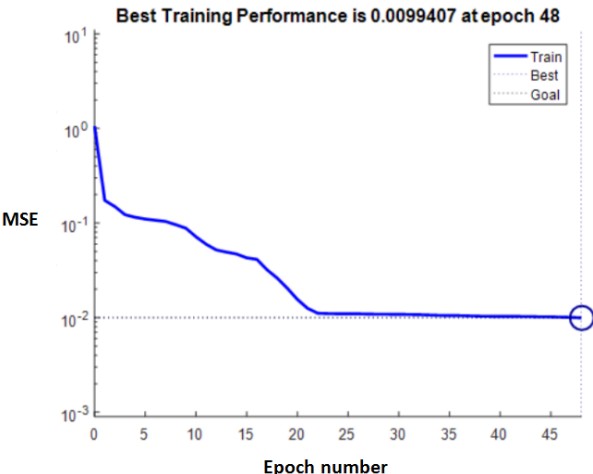

**Figure 5.** BP neural network training performance.

**Table 5.** Comparison of the predicted and actual output values.

| Project | Network Output | | | Expected Output | | | Output Signal Matching Achieved |
|---|---|---|---|---|---|---|---|
| 1 | 0 | 1 | 0 | −0.0088 | 0.8482 | 0.1606 | yes |
| 2 | 0 | 0 | 1 | −0.0103 | 0.3404 | 0.6699 | yes |
| 3 | 1 | 0 | 0 | 0.3799 | 0.2096 | 0.4106 | no |
| 4 | 0 | 1 | 0 | 0.4959 | −0.3353 | 0.8394 | no |
| 5 | 0 | 0 | 1 | −0.0444 | 0.0245 | 1.0199 | yes |
| 6 | 0 | 0 | 1 | −0.0711 | 0.2571 | 0.8140 | yes |
| 7 | 1 | 0 | 0 | 0.0665 | 0.8324 | 0.1011 | no |
| 8 | 0 | 1 | 0 | −0.4850 | 0.8688 | 0.6163 | no |
| 9 | 0 | 0 | 1 | −0.3857 | 0.4871 | 0.8986 | yes |
| 10 | 1 | 0 | 0 | 0.2367 | 0.8477 | −0.0843 | no |

## 6. Conclusions

In this study, the software project risk assessment model based on the rough set and BP neural network was proposed. By combining the advantages of these two algorithms, the risk prediction problem was solved under the limitation of data incompleteness and inaccuracy.

Using the methods presented in the related literature, expert interview, and brainstorming, the initial list of risk factors of the software project was constructed. According to this list, questionnaires were distributed to software project managers and software development engineers. The training samples and test samples were obtained from 91 questionnaire responses. First, the rough set software was adopted to perform attribute reduction to obtain the learning samples, and then, the obtained learning samples were used to train the BP neural network assessment model. Moreover, the test sample set was used to test the trained network model. The test results indicated that, compared with the single BP neural network model, the classification and prediction accuracy of the proposed model based on the combination of the rough set and BP neural network was higher. Hence, the validity of the proposed joint model was verified both theoretically and empirically. With a better software project risk assessment model, threats to successful operations and business management are easier to be identified, addressed, and eliminated for better enterprise performance.

The shortcoming of the proposed model is that the score of risk factors relied on expert experience; thus, the individual subjectivity was strong. In our future work on the risk assessment, the objective risk data will be obtained from the software project risk management system by virtue of big data technology as correctly as possible to enhance the objectivity of a sample set.

**Author Contributions:** X.L. conceived and designed the conceptualization, formal analysis and methodology, and performed writing; Q.J. performed the data curation, fund acquisition, and writing—review & editing; M.H. being supervision and validation; Q.C. took care of the data, software, and writing.

**Funding:** This work is supported by the Fujian Soft Science Research Plan Project (grant number 2019R0093), Social Science Foundation of Fujian (grant number FJ2018B062 & FJ2019B101) and the Xiamen Science and Technology Plan Project (grant number 2018S2247).

**Conflicts of Interest:** The authors declare no conflict of interest.

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
