# Peer review of "Support or Risk? Software Project Risk Assessment Model Based on Rough Set Theory and Backpropagation Neural Network"

_sustainability, doi:10.3390/su11174513_

Round 1

Reviewer 1 Report

The subject of the article is very interesting. Literature review can be improved with more recent research. The representatives of the sample is not mentioned. Also, the methods of experts selection is not mentioned. 

In related works section only two references are after 2010, the rest of them are older, especially from “90. At line 272 is Rosetta, not Rostta. I recommend to authors to justify why they selected 10 project and if they used some criteria for their selection. I also recommend to justify why they consider that the number of experts is representative for their research.

Author Response

Thank you for your concern!

Reviewer 2 Report

The article entitled Support or Risk? Software Project Risk Assessment Model based on Rough Set Theory and Backpropagation Neural Network is well written and from my point of view would be of interest for the readers of sustainability. In spite of these, and before its publication, I would like to recommend some minor changes that would be of interest in order to improve the average quality of the paper. These changes are as follows:

Author speaks about sucess in software engineering projects. I suggest comparing the sucess rate with other industries in order to know if it is high or not. Lines 38 and 39: please check the right spelling of Bohem or Boehm here and in the bibliographical references also. From line 121: formulae are employed in the text but they does not seem to be properly inserted. The section about back propagation neural networks starts in line 148. Please note that there is no bibliographical references in the whole section. How is it possible? Please check again this section.

Author Response

Thank you for your concern!
